# Functional Impairment of Endothelial Colony Forming Cells (ECFC) in Patients with Severe Atherosclerotic Cardiovascular Disease (ASCVD)

**DOI:** 10.3390/ijms23168969

**Published:** 2022-08-11

**Authors:** Stéphanie Simoncini, Simon Toupance, Carlos Labat, Sylvie Gautier, Chloé Dumoulin, Laurent Arnaud, Maria G. Stathopoulou, Sophie Visvikis-Siest, Pascal M. Rossi, Athanase Benetos, Françoise Dignat-George, Florence Sabatier

**Affiliations:** 1Aix Marseille Univ, INSERM, INRAE, C2VN, 13005 Marseille, France; 2Université de Lorraine, INSERM, DCAC, 54000 Nancy, France; 3Université de Lorraine, CHRU-Nancy, Department of Geriatrics, 54000 Nancy, France; 4Cell Therapy Laboratory, Hôpital de la Conception, AP-HM, INSERM CIC BT 1409, 13005 Marseille, France; 5Hematology and Vascular Biology Department, Hopital de la Timone, AP-HM, 13005 Marseille, France; 6INSERM, C3M, Team Control of Gene Expression (10), Université Cote d’Azur, 06000 Nice, France; 7Université de Lorraine, IGE-PCV, 54000 Nancy, France; 8Service de Médecine Interne, Hôpital Nord, APHM, 13015 Marseille, France

**Keywords:** endothelial progenitor cells, atherosclerosis, senescence

## Abstract

Endothelial dysfunction is a key factor in atherosclerosis. However, the link between endothelial repair and severity of atherosclerotic cardiovascular disease (ASCVD) is unclear. This study investigates the relationship between ASCVD, markers of inflammation, and circulating endothelial progenitor cells, namely hematopoietic cells with paracrine angiogenic activity and endothelial colony forming cells (ECFC). Two hundred and forty-three subjects from the TELARTA study were classified according to the presence of clinical atherosclerotic disease. ASCVD severity was assessed by the number of involved vascular territories. Flow cytometry was used to numerate circulating progenitor cells (PC) expressing CD34 and those co-expressing CD45, CD34, and KDR. Peripheral blood mononuclear cells ex vivo culture methods were used to determine ECFC and Colony Forming Unit- endothelial cells (CFU-EC). The ECFC subpopulation was analyzed for proliferation, senescence, and vasculogenic properties. Plasma levels of IL-6 and VEGF-A were measured using Cytokine Array. Despite an increased number of circulating precursors in ASCVD patients, ASCVD impaired the colony forming capacity and the angiogenic properties of ECFC in a severity-dependent manner. Alteration of ECFC was associated with increased senescent phenotype and IL-6 levels. Our study demonstrates a decrease in ECFC repair capacity according to ASCVD severity in an inflammatory and senescence-associated secretory phenotype context.

## 1. Introduction

Atherosclerotic cardiovascular disease (ASCVD) development is a multifactorial process implying chronic inflammation and vascular remodeling associated with the disruption of the functional and structural integrity of the endothelium. Common conditions predisposing to atherosclerosis, such as hypertension, diabetes, hyperlipidemia, and smoking, as well as premature or accelerated aging, are associated with endothelial dysfunction [1,2,3]. Onset of ASCVD is therefore dependent on the balance between injury and capacity for repair of the endothelial monolayer. Endothelial repair is accomplished by the migration of surrounding mature endothelial cells, but these terminally differentiated cells have low proliferative potential and limited ability to replace the damaged endothelium. Accumulating evidence has demonstrated that endothelial progenitor cells (EPC), besides being involved in vessel growth and angiogenesis, could facilitate the reendothelialization of injured arteries by replacing dysfunctional endothelial cells or secreting protective paracrine factors, thereby limiting the progression of vascular damage. Consistently, pivotal studies have established a relationship between low circulating EPC count or activity and progression of ASCVD [4,5] and identified altered EPC levels as a relevant biomarker of adverse cardiovascular outcome [6]. Some studies also demonstrated the early prognostic value of EPC monitoring in cardiovascular disease [7,8]. However, these studies often used a poorly defined label “EPC” for addressing heterogeneous cell types displaying repair capacity, and this critically hampers the appraisal of how these cells may contribute to the maintenance of vascular homeostasis. Indeed, a recent international consensus statement established that the majority of cells previously defined as EPC based on a CD34, KDR, CD133 phenotype or colony forming unit-endothelial cells rather correspond to myeloid cells with paracrine angiogenic activity. By contrast, endothelial colony forming cells (ECFC) behave as relevant progenitor cells of the endothelial and non-hematopoietic lineage, mainly originating from the vascular wall stem cell niche [9] and displaying a strong vasculogenic and repair activity. However, modulation of ECFC under cardiovascular risk factors and their role in cardiovascular risk genesis remain poorly investigated, in part due to the heavy ex vivo culture methods needed to characterize ECFC in laboratory practice. Yet, reduced ECFC-mediated vascular repair functionality is believed to contribute to an increased risk of coronary artery disease [10], and increased senescence of ECFC is emerging as an important mechanism compromising endothelial repair in aging and cardiovascular situations [11]. Moreover, it is now admitted that ECFC can be considered as potential liquid biopsies in several vascular disorders [12,13,14].

In addition to EPC, circulating inflammatory molecules are recognized as fundamental players in ASCVD [15]. Among them, the pro-inflammatory cytokine interleukine-6 (IL-6), is considered as a serum marker of atherosclerosis as well as a major component of senescence-associated secretory phenotype (SASP) [16,17,18,19]. The proangiogenic cytokine vascular endothelial growth factor (VEGF-A), a key mediator of angiogenesis, is also playing a role in endothelial repair and inflammation [20,21] and may have an impact on endothelial progenitors’ functions [22].

The present study aimed to define the relationship between circulating biomarkers of endothelial repair, namely hematopoietic cells with paracrine angiogenic activity and ECFC, markers of inflammation, and the presence and severity of atherosclerotic disease. By providing the first endothelial status integrative approach including the phenotypic and functional characterization of ECFCs in the context of atherosclerotic disease, we expected to provide refined appraisal of the potential role of ECFC in the pathogenesis of atherosclerosis. 

## 2. Results

### 2.1. Characteristics of the Population 

Clinical characteristics of the participants are shown in Table 1. Subjects with ASCVD were older, more likely to be male, and showed a higher prevalence of cardiovascular risk factors, such as hypertension, diabetes, dyslipidemia, and tobacco smoking. Subjects in ASCVD group had a significantly higher carotid thickness and displayed more carotid plaques than controls. In addition, their pulse wave velocity (PWV) and pulse pressure (PP) were higher while their diastolic blood pressure (DBP) was lower than controls. The level of circulating IL-6 was increased in the plasma of ASCVD subjects and in subjects with presence of carotid plaque (without plaques; 1.00 (0.72–2.72) vs. with plaque; 2.78 (1.15–5.92) pg/mL; *p* < 0.001). Moreover, IL-6 level displayed a positive association with the number of territories with ASCVD (trend *p* < 0.001). On the contrary, circulating VEGF was reduced in the plasma of ASCVD subjects and was negatively associated with number of ASCVD sites (trend ANOVA *p* < 0.001). No significant difference was observed in mean BMI values between ASCVD patients and controls.

According to the group classification, there were significant differences in the use of antihypertensives, anti-diabetics, anti-coagulant drugs, and statins between ASCVD patients and controls (Table 1, Treatments). In addition, we observed that ASCVD patients took a mean of 4.7 drugs, including 4.4 CV and metabolic drugs. The two most frequently taken substance classes were anti-hypertensive drugs and statins. Anti-coagulant drugs were taken by 17% of ASCVD patients vs. 7% of the controls. 

### 2.2. Characterization of EPCs in Patients with ASCVD

No difference was observed in the total level of circulating CD34+ cells (Table 2A) between ASCVD patients and controls. However, cytometry analysis revealed that CD45+CD34+KDR+ cells count was significantly higher in ASCVD patients compared to controls (Table 2A, *p* = 0.02) and was related to increased ASCVD severity defined by the number of ASCVD sites (Table 2B, *p* = 0.004). These results persisted after adjustment for age and sex. No significant modulation was observed for the CFU-EC subset despite a trend of an increased number in the blood of patients with the most severe ASCVD (*p* = 0.09 after adjustment for age and sex).

The percentage of ECFC obtained from peripheral blood mononuclear cells (PBMC) was not different between ASCVD patients and controls (Figure 1A) but decreased significantly with the ASCVD severity expressed by the number of ASCVD sites (*p* = 0.03, Figure 1B). Similarly, ECFC number (number of colonies/millions of PBMC) was not different between ASCVD patients and controls (Figure 1C) but we observed a significant decrease in patients with ASCVD affecting more than one territory (*p* = 0.03, Figure 1D). Taken together, these results indicate that despite an increased number of cells with a precursor phenotype in whole blood, the capacity of ECFC to form colonies was reduced according to ASCVD severity.

### 2.3. ECFC Functionality According to the ASCVD Presence and Severity

ASCVD patients showed a lower ECFC proliferation (Figure 2A, *p* = 0.04), this proliferation defect being more marked in ECFC from patients with ASCVD in more than one territory (data not shown). Reduced growing capacity was associated with morphological changes and appearance of enlarged flattened cells among ECFC from ASCVD patients (Figure 2B). Using SA-β-galactosidase staining, we found that ASCVD patients showed a higher ECFC senescence (Figure 2B,C, *p* = 0.02), especially in the case of more than one territory of ASCVD (data not shown).

In a 3D spheroid assay, we observed an alteration in the vasculogenic capacities of ECFC from ASCVD patients (Figure 3A) expressed by fewer branching points/spheroid than controls (Figure 3B, *p* = 0.04) without any modulation of the sprout number/spheroid (Figure 3C) or the cumulative length (Figure 3D). However, we observed a clear reduction in the sprout number, branching points, as well as cumulative length for the patient presenting ASCVD in more than one territory (data not shown).

Interestingly, senescence was inversely correlated with angiogenic capacities of ECFC (Figure 4A–C). Indeed, the senescence was negatively correlated with the number of sprouts per spheroid (Figure 4A, *p* = 0.016) and trends were observed with the number of branching points (Figure 4B) and the cumulative length (Figure 4C). All these findings indicate that ASCVD impairs the in vitro angiogenic properties of ECFC in a severity-dependent manner with the strongest reduction of angiogenic properties being observed in subjects with higher number of ASCVD sites.

### 2.4. Associations between PC Subpopulations Counts and Functionality, Arterial Phenotypes, and Circulating Markers 

No association was observed between blood pressure or PWV with cellular quantification or phenotypes. However, CD45+CD34+KDR+ cell counts were negatively correlated with carotid thickness (*p* = 0.017; data not shown). In addition, a trend was observed for a negative association between ECFC counts and the presence of carotid plaque (*p* = 0.087; data not shown). We did not find any association for circulating CD34+ cells and CFU-EC counts with the arterial phenotypes and circulating soluble markers.

Senescence of ECFC was positively correlated with carotid thickness (*p* = 0.013; Figure 5B) and proliferative capacity of ECFC was inversely associated with IL-6 levels (*p* = 0.03; Figure 5C). These associations persisted after adjustment for age and sex. In addition, we observed a positive association between SBP, PP, and PWV with IL-6 levels (*p* < 0.05). These associations disappeared after adjustment for age and sex.

No association was observed between VEGF levels and proliferative capacity or senescence of ECFC (Figure 5E,F). However, we highlighted a positive correlation between circulating VEGF levels and the angiogenic properties of ECFC (Figure 6). A positive correlation between VEGF levels and cumulative germination and branching length (*p* = 0.01) was found in non-statin treated subjects (*n* = 38; *p* = 0.002) but no correlation was observed in treated patients (*n* = 17; *p* = 0.32; VEGF statin interaction *p* = 0.01).

## 3. Discussion

In the present study, we evaluated the link between endothelial repair status and the severity of the atherosclerotic disease. We found that: (1) despite an increased number of precursors circulating in whole blood, the capacity of ECFC to form colonies was altered according to ASCVD severity. (2) ASCVD impaired in vitro angiogenic properties of ECFC in a severity-dependent manner with the strongest reduction in angiogenic properties observed in subjects with ASCVD in more than one territory. (3) Functional alteration of ECFC was linked to an increased senescent phenotype and an inflammatory environment.

Many studies have identified circulating EPC as playing an important role in endothelial repair and regeneration. They have been identified as regulators of cardiovascular integrity and involved in the pathogenesis of atherosclerosis when decreased in number and function [5,23,24,25]. However, the current study observed enhanced levels of circulating hematopoietic CD45+CD34+KDR+ and CFU-EC subpopulations without modulation of the CD34+ bone marrow progenitors release, occurring in ASCVD patients depending on the severity. The correlation between CD45+CD34+KDR+ cells levels and the ASCVD status suggests that a vascular injury, such as atherosclerotic lesion, could mobilize the bone marrow-derived pool of progenitor cells displaying a proangiogenic phenotype. In accordance with this hypothesis, recent studies reported EPC mobilization to peripheral blood after acute myocardial infarction [26] or in the context of systemic sclerosis-associated vascular activation [27]. In addition, a recent study reported that factors released from atherosclerotic plaques ex vivo induce EPC mobilization [28]. Because the extent of this mobilization appears to be related to CVD severity, it may represent a crucial mechanism for remodeling and deleterious effects in the inflammatory atherosclerotic context [29,30,31]. Indeed, in ApoE−/− mouse model, Foteinos et al. [32] reported that EPC-treated mice displayed accelerated atherosclerosis along with reduced plaque stability.

Although a higher level of circulating progenitor cells was observed in ASCVD patients, we demonstrate for the first time that isolated ECFC population, which is the vasculogenic subtype of EPC [33], showed fundamental impairments in critical functions such as colony forming ability and proliferation. Indeed, we observed an inverse relationship between ASCVD severity and the number of endothelial colonies. Moreover, the angiogenic impairment of ASCVD ECFC was evidenced by in vitro data showing a reduction in branches number and proliferation capacity of these cells. This reduced functionality was found to be dependent on the ASCVD severity, with a drastic impairment of the angiogenic parameters in the patients with more than one territory affected by ASCVD. Of note, the strength of this result may even be underestimated in our study as difficulty in obtaining clones did not allow for the qualification of ECFC dysfunction in the most affected patients. Consistently, the absence of ECFC colony generation after an ex vivo culture protocol performed according to the internationally standardized methods can be viewed as a hallmark of ECFC depletion or dysfunction in severe ASCVD [33].

Senescence of endothelial progenitor cells limiting the ability for angiogenesis and vascular healing [34,35,36] has emerged as a contributor to endothelial dysfunction developing under cardiovascular disease [37]. We show here for the first time that a senescence profile is found in the ECFC isolated from ASCVD patients. In addition, senescence in ASCVD ECFC is inversely correlated with their functionality, suggesting a key role in the pathophysiological mechanisms driving vascular dysfunction in ASCVD patients. In addition, we observed a significant correlation between ECFC senescence and the carotid thickness. The results reported in the present study agree with other studies demonstrating changes in senescence of endothelial and vascular smooth muscle cells in atherosclerotic lesions [38,39,40,41,42]. Therefore, these results argue in favor of a significant contribution of ECFC accelerated senescence in the poor endothelial regeneration associated to atherogenesis. 

ECFC senescence can be influenced by environmental systemic or local factors as well as by telomere integrity. Interestingly, we recently reported that formation and number of ECFC as well as their replating capacity are associated with telomere length [43]. ASCVD patients display shorter telomere length in leukocytes [44] and this telomere length is correlated with telomere length in ECFC [43]. Then, short telomere length in these cells could explain the observed senescence phenotype and reflect their diminished reparative capacity as previously observed in hematopoietic stem cells [45,46,47]. Besides telomere length, exposure to environmental risks factors such as oxidative stress or inflammation play an important role in ECFC dysfunction as well as in atherosclerosis development [15,48]. In the present study, we observed that a strong increase in IL-6 plasma levels in ASCVD patients inversely correlated with the proliferation capacity of ECFCs. These data are consistent with the literature in which IL-6 is described as a proinflammatory cytokine canonically upregulated in the senescence-associated secretory phenotype of cells that have undergone stress-induced or replicative senescence [49] and proposed as a serum marker of atherosclerosis [15,50]. Indeed, SASP reinforces senescence in an autocrine and paracrine manner, heightens inflammation, and has a detrimental effect on the tissue microenvironment [51]. In addition, we previously described that endothelial microparticles’ release of senescent ECFCs are relevant participants of the ECFC-mediated SAPS and actively contribute to the pathogenesis of cardiovascular disease [52]. By adopting a SASP, ASCVD patients’ cells could produce a series of paracrine factors and inflammatory molecules driving senescence of ECFC, as well as mature endothelial cells and further exacerbate endothelial dysfunction. Altered ECFCs functions mediated by disruption of pro-angiogenic pathway and accelerated senescence may compromise the endothelial repair capacity with significant involvement in the progression of endothelial dysfunction [9,37,53] and development of subsequent diseases such as atherosclerosis. 

Finally, the current study demonstrates a decrease in VEGF plasmatic levels dependent on ASCVD severity and correlated with a decrease of ECFC functional properties. These data are in adequation with the pro-angiogenic role of VEGF [54]. In addition, we observed that the impact of VEGF is partially lost when patients had a history of statin therapy, a parameter known to modulate both VEGF concentrations and the quantity and functionality of EPCs [55,56].

We acknowledge the limitation of our study. The difficulty to obtain replating ECFC clones from ASCVD subjects with high number of ASCVD sites did not allow us to provide a full characterization of angiogenic properties of ECFCs from this group of patients.

In summary, our data shown for the first time a modulation of ECFC quantity and quality according to ASCVD severity in an inflammatory and SASP context. Thus, ECFC functional properties being correlated to disease severity highlight a link between ECFC function and clinical settings, as well as the risk of atherosclerotic events. 

In a clinical perspective, our results demonstrate that ECFC in ASCVD behave as a “liquid biopsy” of the vessel. They could clearly constitute a marker of the clinical vascular phenotype and be a signature of altered endothelial repair capacity prone to favor pro-atherosclerotic conditions. In that way, exploration of ECFC may provide insight into the anticipation of disease progression. Generation of ECFC using a minimally invasive protocol also provides an important tool to determine patient specific endothelial functioning and evaluate or monitor potential treatments with potency to improve vascular regeneration in patients affected or at risk for ASCVD. Therefore, the use of ECFCs provides an attracting vascular disease model to probe mechanisms of endothelial pathogenesis and to delineate therapeutic targets, particularly in conditions associated with adverse environments, such as inflammation and oxidative stress, as in atherosclerosis. This encourages further studies of the molecular mechanisms linking ECFC senescence, alteration of repair capacity and ASCD severity in larger cohort of patients with more than one ASCVD site. In the future, one of the challenges to be addressed will be to explore the possibility of targeting endogenous ECFC and favoring their clonal expansion in an attempt to rescue their regenerative potential.

## 4. Materials and Methods

### 4.1. Subjects

This research draws on the TELARTA (Telomere in Arterial Aging) cohort [57]. Men and women (aged 20 to 94 years) were enrolled in university hospitals in Nancy and Marseille, France. They were admitted for various surgical procedures or for defibrillator implantation. All participants provided written informed consent approved by the Ethics Committee (Comité de Protection des Personnes) of Nancy, France. The study was conducted in accordance with the Declaration of Helsinki and is registered on http://www.clinicaltrials.gov under unique identifier: NCT02176941.

Patients were classified into two groups according to the presence or not of clinical atherosclerosis: ASCVD and Control groups. The ASCVD group included individuals with a history of clinically evident atherosclerosis in at least one of the following vascular territories: heart (coronary arteries), cerebrovascular (carotid and cerebral arteries), and lower limbs (iliac, femoral, and popliteal arteries). The number of territories with ASCVD was calculated considering these three localizations and termed “number of ASCVD”. The patients with more than one site of atherosclerosis were considered as severe ASCVD. 

We excluded subjects with unclear diagnosis as well as subjects for which no exploration of circulating biomarkers had been performed. We excluded 11 subjects with aortic aneurism because the atherosclerotic nature of this arterial disease is debatable (Appendix A).

This analysis is based on a 243 subject subpopulation of the TELARTA cohort in which at least one of the four following circulating biomarkers were explored (Appendix A): circulating levels of CD34 expressing cells, circulating levels of CD45, CD34 and KDR co-expressing cells, Endothelial Colony Forming Cells (ECFC), and Colony Forming Unit-endothelial cells (CFU-EC).

### 4.2. Cells Characterization

Circulating progenitor subpopulations were determined by analyzing the whole circulating progenitor cell (PC) population expressing CD34 and those specifically engaged in the endothelial phenotype defined by co-expression of CD45, CD34, and KDR, using flow cytometry. Peripheral blood mononuclear cells ex vivo culture methods were used to enumerate and analyze Endothelial Colony Forming Cells (ECFC) and Colony Forming Unit- endothelial cells (CFU-EC).

### 4.3. Enumeration and Characterization of Circulating Progenitor Cells by Flow Cytometry

Blood samples were collected for Lithium Heparinate. CD34+ hematopoietic PC were enumerated with a whole-blood flow cytometry protocol adapted from the standardized International Society of Hematotherapy and Graft Engineering single platform sequential gating strategy [58]. Briefly, 100 µL of blood was stained with PE-CD34 antibody or PE-immunoglobulin G1 (IgG1), and 7-aminoactinomycin D (7-AAD) (Stem kit Reagents; Beckman Coulter Life Sciences, Brea, CA, USA), according to the manufacturer’s instructions. After lysis of erythrocytes, flow count beads were added to each sample for absolute value determination, and samples were analyzed with a NAVIOS flow cytometer equipped with CXP software (Beckman Coulter Life Sciences). At least 75,000 cells were acquired by run. CD34+ PCs were identified within 7-AAD-negative viable cells displaying forward scatter (FSC)/side scatter (SSC) characteristics corresponding to the lymphocyte cluster. The PC number in circulation was investigated in 240 subjects. The results were expressed as absolute numbers of CD34 PCs per milliliter of blood.

Because CD34+CD45+KDR+ PCs, identified as circulating angiogenic cells (CACs), are present in low level in peripheral blood, these progenitor subsets were quantified using a four-color flow cytometry strategy after direct immunolabelling of isolated peripheral blood mononuclear cells (PBMC). Only 84 subjects were subject to this quantification due to limitation of blood sample and PBMC number in 157 subjects. PBMCs were isolated by density gradient centrifugation with lymphocyte separation medium (PAA Laboratories, Pasching, Austria) and they were labelled with 10 µL of 7-AAD viability dye, 10 µL of FITC-CD34 antibody (Beckman Coulter), 10 µL of ECD-CD45 antibody (Beckman Coulter), and 10 µL of PE-KDR antibody (R&D Systems, Abingdon, UK). A 10 µL quantity of concentration-matched, PE-conjugated murine IgG1 antibody was used as fluorescence minus one control. After incubation for 20 min at room temperature, cells were washed and re-suspended in 500 µL of PBS, then analyzed using a NAVIOS flow cytometer. 

After selection of 7-AAD-negative cells, KDR cells were identified within CD34+ CD45+ cells displaying FSC/SSC characteristics corresponding to the lymphocyte cluster. At least 5 × 10^5^ viable cells were acquired per run. The percentage of KDR+ cells among CD34+CD45+ cells was determined. Results were expressed as absolute values per milliliter of blood. CD34+CD45+KDR+ PCs value was obtained by multiplying the percentage of each cellular population by the absolute value of CD34+ CD45+HPCs determined as described above.

### 4.4. Assessment of Colony-Forming Unit-Endothelial Cells

Clonogenic assays allowing the enumeration of Colony Forming Unit- endothelial cells (CFU-EC) after ex vivo culture of peripheral blood mononuclear cells have been performed according to the method described previously by Hill et al. [4], and adapted by Smadja et al. [59]. PBMC were obtained by density gradient isolation and cultured with the Endocult^®^ Liquid Medium Kit (STEMCELL Technologies, Vancouver, BC, Canada) according to the manufacturer’s instructions. Briefly, PBMC were re-suspended in complete Endocult^®^ medium and seeded at 5 × 10^6^ cells/well in fibronectin-coated tissue culture plates (BD Biosciences, San Jose, CA, USA). After 48 h, to obtain CFU-ECs, non-adherent cells were collected and plated in Endocult^®^ medium at 10^6^ cells/well in 24-well fibronectin-coated plates. CFU-EC colonies were counted after another 3 days, as recommended by the manufacturer. This assessment was performed in 220 subjects.

### 4.5. Assessment of Endothelial Colony Forming Cells

Determination of Endothelial Colony Forming Cells (ECFC), also called late EPC, was performed according to the culture method reported previously [60]. Briefly, PBMC were plated on gelatin-coated plates in EGM-2MV culture medium. ECFC appeared as adherent cells with typical cobblestone morphology after 10–22 days of culture. ECFC were enumerated by visual inspection using an invert microscope (Leica) under 5× magnification. A monolayer containing at least 20 cells with cobblestone pattern was counted as a colony.

The exploration of the ECFC subpopulation has been performed in 112 subjects and primary colonies obtained in 86 of them. 

### 4.6. ECFC Functional Tests (Proliferation, Senescence, and Vasculogenic Properties)

Endothelial colonies of ECFC obtained from culture of PBMC were used in in vitro assays aiming to determine the functionality of these cells. Primary colonies were detached from culture plates by trypsin/EDTA treatment and serially expanded in EGM2-MV culture medium until passage 3. Among the 86 primary colonies obtained from the 112 patients, only 58 showed replating efficiency and were expanded in sufficient quantity to allow the realization of functional tests. Isolated ECFC were then analyzed for proliferation capacity using BrdU incorporation assay (Roche Molecular Biochemicals, Mannheim, Germany) according to the manufacturer’s instructions. Senescence-associated-ß-galactosidase (SA-ß-gal) activity was performed using a senescence detection kit (BioVision Research Products, Waltham, MA, USA) according to the manufacturer’s instructions. Percentage of SA ß-gal positive cells was counted in 10 randomly selected microscopic fields (magnification 20×). 

ECFC vasculogenic potential was determined using a spheroid assay as described by Korff et al. Briefly, ECFC were suspended in culture medium containing 0.2% (*w/v*) carboxymethylcellulose and seeded in non-adherent round-bottomed wells at a density of 400 cells/well. Spheroids were generated overnight and then collected and embedded into collagen I gel. The spheroid containing gel was rapidly transferred into pre-warmed Labtek II slides and allowed to polymerize (30 min), and then 100 µL of EGM2-MV was added on top of the gel. Following 24 h of culture, the spheroids were fixed for 30 min in 4% paraformaldehyde at room temperature. After washing, pictures were taken with a phase-contrast Leica DMI8 microscope. To measure the cumulative sprout length per spheroid, every sprout from 20 spheroids was assessed, and the mean cumulative sprout length per spheroid was calculated. In addition, the number of branch points and the number of sprouts were counted from 20 spheroids and calculated as mean number of branch points and mean number of sprouts per spheroid.

These functional tests were quantified in 55 subjects. In the three remaining subjects, a functional test was not obtained for technical reasons.

### 4.7. IL-6 and VEGF Plasmatic Levels Determination

IL-6 and VEGF-A proteins were measured in plasma samples using Cytokine Array I on Randox semi-automated benchtop immunoanalyzer (Evidence Investigator Analyzer, Randox Laboratories Ltd., Crumlin, UK). Cytokine Array I is a high sensitivity multiplex cytokine and growth factor array that enables simultaneous detection of 12 cytokines and growth factors in a single sample.

### 4.8. Characterization of Arterial Phenotype

#### 4.8.1. Carotid Femoral Pulse Wave Velocity (PWV)

PWV was assessed with the Sphygmocor^®^ (AtCor Medical, Sydney, Australia) device [61]. To conduct a PWV measurement, a blood pressure cuff is placed around the femoral artery of the patient to capture the femoral waveform, and a tonometer pressure sensor is used to capture the carotid waveform. The pulses are collected over a pre-set time, the pulse transit time which is the time that the pulse takes to travel from the carotid artery to the femoral artery. The distance between the carotid and femoral arteries is measured, and the pulse wave velocity automatically determined by dividing the distance by the pulse transit time.

#### 4.8.2. Carotid Artery Intima-Media Thickness (IMT) and Presence of Carotid Atherosclerotic Plaque

The intima-media thickness and presence of carotid atherosclerotic plaque were measured using the high-resolution B mode ultrasound method. Measurements were performed on the common carotid (right and left), on a 2–4 cm segment predefined by the observer, at 2–3 cm from the carotid bifurcation in B-mode ultrasound as previously reported. Briefly, measurements were performed along the longitudinal axis of the artery and comprised an associated diameter measurement calculated automatically or semi-automatically using an image analysis software. It allowed changes in distance between the two interfaces several times smaller than the axial resolution of the ultrasound probe (depending on software used) to be determined. Image acquisition was performed visually on an ultrasound with a 7.5 MHz probe.

### 4.9. Measurement of Blood Pressure

Arterial blood pressure was recorded before and after the ultrasound examination, using an automatic device (Omron 705IT, Omron Co., Kyoto, Japan) at a frequency of one measurement every 2 min. Three consecutive measurements were carried out after 10 min rest in supine position. Pulse pressure was calculated as the difference between systolic and diastolic pressure.

### 4.10. Statistical Analysis

Continuous variables are presented as means (SD) or median (interquartile range) for normally and non-normally distributed data, respectively. Discrete variables are presented as percentages. Comparisons were performed using the Mann–Whitney or Khi2-test as appropriate. ANOVA trend tests were used for the analysis of number of ASCVD. Adjustments for age and sex were made by using a general linear model after normalization of non-normally distributed variables. Bivariate relationships between continuous variables were determined using Pearson and Spearman correlation coefficients as appropriate. A *p* < 0.05 was considered significant. Statistical analyses were performed using the NCSS 9 statistical software package (NCSS, Kaysville, UT, USA).

## Figures and Tables

**Figure 1 ijms-23-08969-f001:**
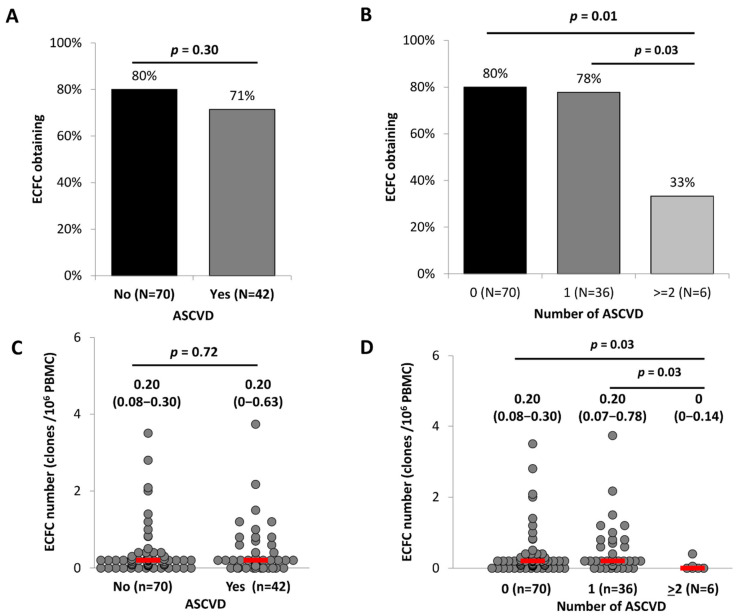
**ECFC formation according to ACSVD presence and severity**. PBMC from 112 patients were seeded onto gelatin-coated wells and observed every 2 days as described in Materials and Methods. (**A**,**B**) Colony progeny obtention (in%) in the presence or absence of ASCVD (**A**) or according to the extent of ASCVD (**B**). (**C**,**D**) Number of ECFC colonies formed by PBMC according to the presence or absence of ASCVD (**C**) or the extent of ASCVD (**D**). Data are presented as median (IQR) and percentages (%). ASCVD indicates atherosclerotic cardiovascular disease. *p* of comparisons are obtained from the Mann–Whitney or Khi2-test as appropriate.

**Figure 2 ijms-23-08969-f002:**
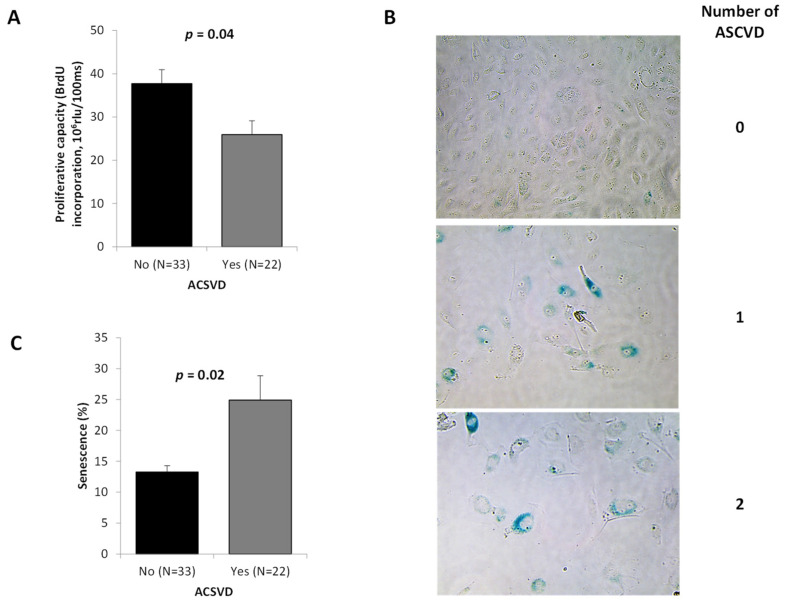
**ECFC proliferative capacity and senescence according to ASCVD presence and severity.** ECFC were used at passage 3 as described in Materials and Methods. (**A**) Proliferation assayed by BrdU incorporation. Results were expressed in rlu/s of luminescence measurements. Data represent mean ± SD of 55 independent experiments performed in triplicate. (**B**) Representative images from ECFC according to the extent of ASCVD (magnification 20×). (**C**) The percentage of senescent cells was determined as the number of cells expressing SA-β-galactosidase (blue cells) relative to the total number of cells in each field. Histograms represent mean ± SD of 55 independent experiments performed in triplicate. ASCVD indicates atherosclerotic cardiovascular disease. *p* of comparisons are obtained from the Mann–Whitney test.

**Figure 3 ijms-23-08969-f003:**
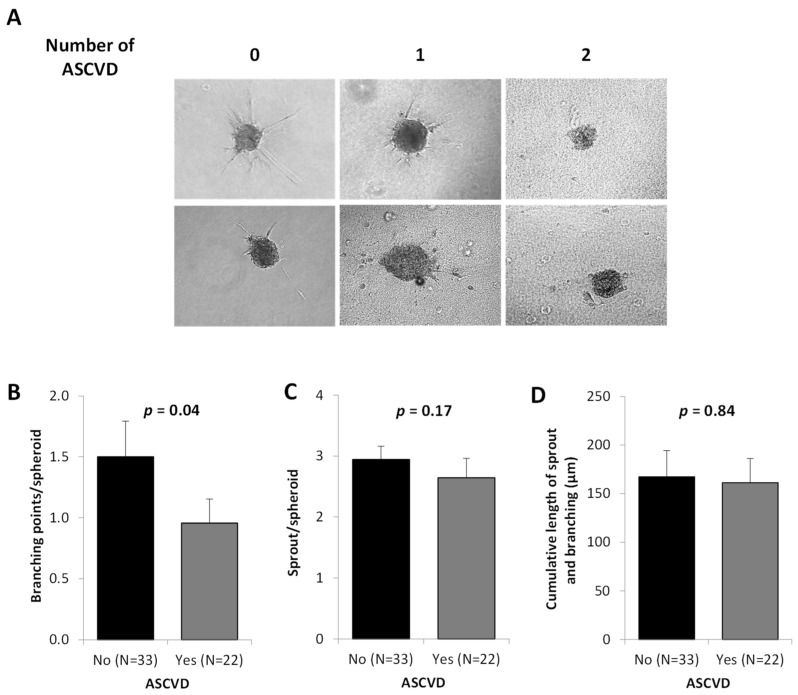
**ECFC vasculogenic potential according to the presence and severity of ASCVD.** ECFC were used at passage 3 as described in Materials and Methods. (**A**) Representative experiment of capillary-like sprout formation from 3D in vitro angiogenesis assay with collagen gel-embedded spheroids obtained with ECFC from patients according to the extent of ASCVD (number of ASCVD) (magnification 20×). (**B**–**D**) Quantification of the number of branching points (**B**), the number of sprouts (**C**), and the cumulative length (**D**) per spheroids according to the absence or presence of ASCVD. For each experiment, sprouts from 20 spheroids were counted. Data represent means ± SD of 55 independent experiments. ASCVD indicates atherosclerotic cardiovascular disease. *p* of comparisons are obtained from the Mann–Whitney test.

**Figure 4 ijms-23-08969-f004:**
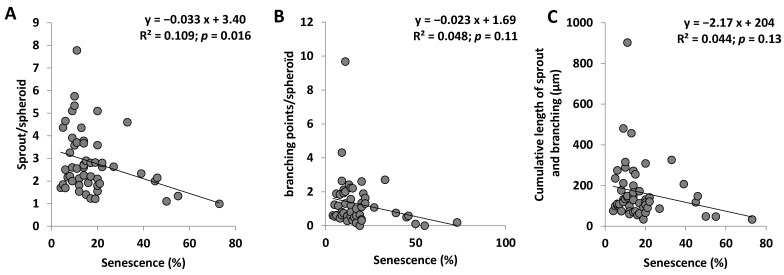
**Relation between senescence and angiogenic functions of ECFC.** (**A**) Senescence according to the number of sprouts per spheroid. (**B**) Senescence according to the number of branching points per spheroid. (**C**) Senescence according to the cumulative length per spheroid. R^2^ and *p* are from Pearson correlation.

**Figure 5 ijms-23-08969-f005:**
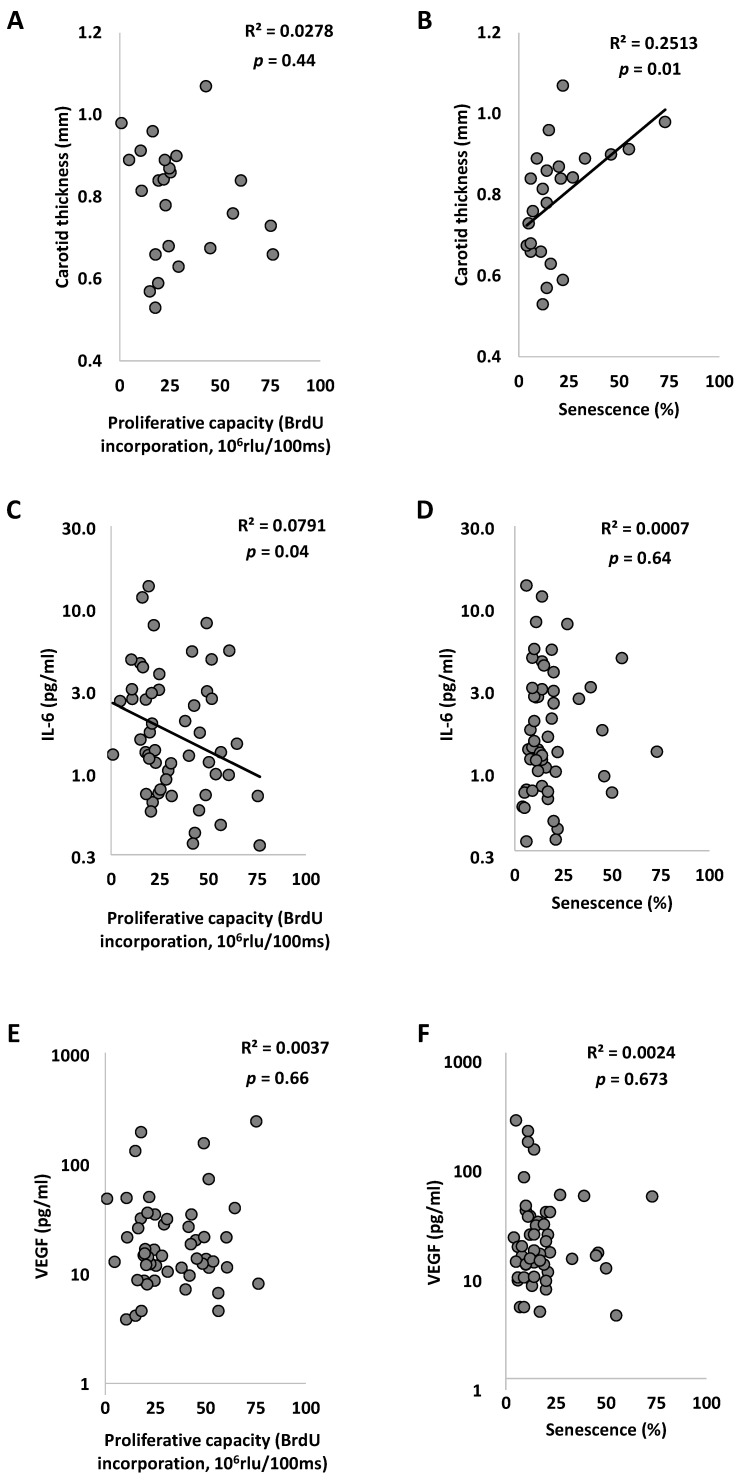
**Relations between ECFC proliferative capacity or senescence and clinical characteristics.** (**A**,**B**) Carotid thickness (mm) according to the proliferative capacity and the senescence of ECFC. (**C**,**D**) IL-6 level according to the proliferative capacity and the senescence of ECFC. (**E**,**F**) VEGF level according to the proliferative capacity and the senescence of ECFC. R^2^ and *p* are from Pearson correlation. IL-6 indicates Interleukin 6 and VEGF indicates vascular endothelium growth factor.

**Figure 6 ijms-23-08969-f006:**
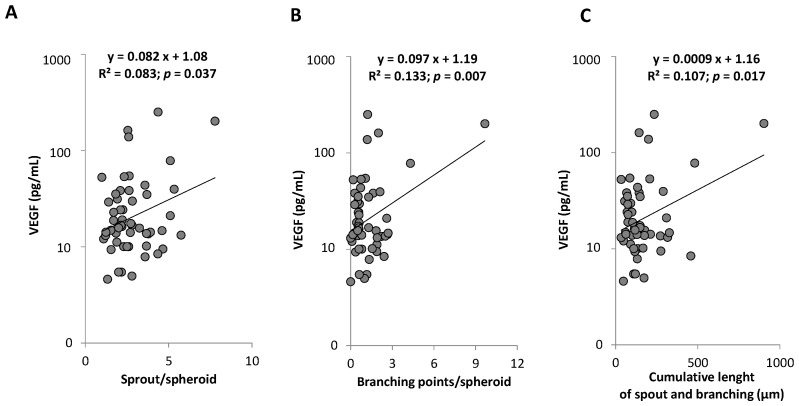
**Relation between VEGF level and angiogenic function of ECFC.** (**A**) VEGF level according to the number of sprouts per spheroid (**B**) VEGF level according to the number of branching points per spheroid (**C**) VEGF level according to the cumulative length per spheroid. R^2^ and *p* are from Pearson correlation. VEGF indicates vascular endothelium growth factor.

**Table 1 ijms-23-08969-t001:** **Clinical characteristics of participants.** Data are presented as mean ± SD, median (IQR), or percentages (%).

	All Participants	N	ASCVD			
			No	N	Yes	N
Gender (women%)	35%	243	52%	124	17% ***	119
Age (years)	62 ± 14	243	57 ± 16	124	68 ± 10 ***	119
BMI (kg/m^2^)	26.9 ± 6.1	240	27.0 ± 5.8	124	26.8 ± 6.4	116
**ASCVD sites**						
Coronary	24%	243	-	124	49%	119
Carotido-cerebral	16%	243	-	124	32%	119
Femoro-popliteal	23%	243	-	124	47%	119
Others	2%	243	-	124	3%	119
Number of ASCVD sites	0.64 ± 0.74	243	-	124	1.31 ± 0.50	119
ACSVD	49%	243	-	124	100%	119
**ASCVD Risk factors**						
Hypertension	45%	242	34%	123	57% ***	119
Diabetes	18%	242	10%	123	27% ***	119
Dyslipidaemia	23%	242	12%	123	34% ***	119
Smoking	50%	242	29%	123	71% ***	119
**Treatments**						
Anti-hypertensive drugs	59%	243	40%	124	79% ***	119
Anti-diabetic drugs	15%	243	7%	124	24% ***	119
Anti-coagulant drugs	12%	243	7%	124	17% *	119
Statins	41%	243	15%	124	68% ***	119
Number of CV and metabolic drugs	2.9 ± 2.7	243	1.4 ± 1.9	124	4.4 ± 2.5 ***	119
Total number of drugs	3.1 ± 2.8	243	1.6 ± 2.0	124	4.7 ± 2.6 ***	119
**Blood pressure**						
SBP (mmHg)	135 ± 18	179	133 ± 17	95	136 ± 20	84
DBP (mmHg)	76 ± 10	179	78 ± 10	95	74 ± 10 **	84
MBP (mmHg)	95 ± 11	179	96 ± 11	95	94 ± 11	84
PP (mmHg)	59 ± 16	179	55 ± 14	95	63 ± 17 **	84
HR (bpm)	73 ± 14	179	75 ± 13	95	72 ± 15	84
**Arterial phenotypes**						
PWV (m/s)	12.30 ± 2.74	169	11.77 ± 2.50	89	12.88 ± 2.89 *	80
Carotid thickness (mm)	0.79 ± 0.18	129	0.74 ± 0.18	71	0.85 ± 0.15 ***	58
Carotid plaque presence	65%	127	51%	69	83% ***	58
**Plasmatic markers**						
IL-6 (pg/mL)	1.90 (0.95–4.78)	237	1.16 (0.71–2.76)	120	3.13 (1.36–6.48) ***	117
VEGF (pg/mL)	15.7 (10.1–27.0)	237	17.3 (12.26–30.89)	120	14.06 (7.32–23.08) ***	117

No and Yes indicate absence or presence of ASCVD, respectively. ASCVD indicates atherosclerotic cardiovascular disease; BMI, body mass index; SBP, systolic blood pressure; DBP, diastolic blood pressure; HR, hear rate; MBP, mean blood pressure; PP, pulse pressure; PWV, pulse wave velocity; IL-6, interleukin 6; VEGF, vascular endothelium growth factor. * *p* < 0.05; ** *p* < 0.01; *** *p* < 0.001. Non-parametric test (Mann–Whitney or Khi2) vs. No ASCVD.

**Table 2 ijms-23-08969-t002:** **Circulating progenitor cell subpopulations according to ASCVD presence and severity**. Enumeration and characterization of circulating progenitor cells (CD34+, CD45+CD34+KDR+) by flow cytometry and assessment of colony forming unit-endothelial cells (CFU-EC) in all patients included in the study. Results are expressed as the median (IQR). (**A**) Levels of progenitor’s subsets in the presence or absence of ASCVD. No and Yes indicate absence or presence of ASCVD, respectively. (**B**) Levels of progenitor’s subsets in patients classified according to the extent of ASCVD (Number of ASCVD sites).

**(A)**	**All Participants**	**N**	**ASCVD**				** *p* **
			**No**	**N**	**Yes**	**N**	
**CD34+ cells (/mL)**	1324 (802–2023)	240	1338 (880–2032)	122	1297 (722–2021)	118	0.75
**CD45+CD34+KDR+ (/mL)**	8.09 (0–24.85)	84	0 (0–20.56)	33	13.68 (0–31.84)	51	0.02
**CFU-EC (/million cells)**	0.49 (0–3.45)	220	0.40 (0–3.27)	112	0.51 (0–3.80)	108	0.68
**(B)**	**Number of ASCVD Sites**						**Trend** ** *p* **
	**0**	**N**	**1**	**N**	**>2**	**N**	
**CD34+ cells (/mL)**	1338 (880–2032)	122	1256 (701–2025)	83	1423 (1036–1846)	35	0.93
**CD45+CD34+KDR+ (/mL)**	0 (0–20.56)	33	11.56 (0–23.53)	32	22.79 (5.55–37.70)	19	0.004
**CFU-EC (/million cells)**	0.40 (0–3.27)	112	0.20 (0–3.06)	75	1.29 (0–9.99)	33	0.33

ASCVD indicates atherosclerotic cardiovascular disease. *p*: non-parametric test (Mann–Whitney) between No and Yes groups; trend *p*: ANOVA trend test between the 3 groups.

## Data Availability

The data presented in this study are available from the corresponding author on reasonable request.

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
