# Peer review of "Functional Impairment of Endothelial Colony Forming Cells (ECFC) in Patients with Severe Atherosclerotic Cardiovascular Disease (ASCVD)"

_ijms, 2022, doi:10.3390/ijms23168969_

Round 1

Reviewer 1 Report

Dear Editor,

I evaluated this interesting paper from Simoncini et al. Some remarks should be addressed:

1. Inclusion and exclusion criteria should be better specified and described for this paper. Please revise the entire work.

2. I think that a flow chart of the study would improve the comprehension of the study design. Please provide.

3. Please revise some typos in the text.

4. Authors wrote "severe ASCVD": what did the term "severe" stand for? Please specify.

5. Pharmacological treatments should be better describe as they can impact on results.

6. How many patients were on atrial fibrillation? and how many patients were on anticoagulation therapy? This should be addressed.

Author Response

We thank the reviewer for these remarks and these interesting comments.

  1. Inclusion and exclusion criteria should be better specified and described for this paper. Please revise the entire work.

We reformulated the part of the Methods section regarding the inclusion and exclusion criteria to improve clarity (line 327 to 330).

  1. I think that a flow chart of the study would improve the comprehension of the study design. Please provide.

We realized two flow charts of the study design that we included as supplemental data and mentioned in the Methods section lines (344 and 348). 

  1. Please revise some typos in the text.

To take this remark into account, we have carried out a careful re-reading of the manuscript.

  1. Authors wrote "severe ASCVD": what did the term "severe" stand for? Please specify.

We agree with the reviewer’s comment. We have included a sentence in line 325 to specify what did the term “severe” stand for.

  1. Pharmacological treatments should be better described as they can impact on results.

We included data on pharmacological treatments in revised table 1 and shortly discussed the results in lines 99 to 104.

  1. How many patients were on atrial fibrillation? and how many patients were on anticoagulation therapy? This should be addressed.

We did not have access to data on atrial fibrillation in this cohort but we included the available data on anticoagulation therapy in the revised table 1 and discussed the result line 103.

Reviewer 2 Report

The paper, entitled "Functional impairment of endothelial colony forming cells (ECFC) in patients with severe atherosclerotic cardiovascular disease (ASCVD)” aims to investigates the relations between ASCVD, markers of inflammation, and circulating endothelial progenitor cells, namely hematopoietic cells with paracrine angiogenic activity and endothelial colony forming cells (ECFC). For fill the main goal, 243 subjects from the TELARTA study were classified according to the presence of clinical atherosclerotic disease. It was found that alteration of ECFC were associated with increased senescent phenotype and IL-6 levels. Two points came out:

-        How can be translated to clinical setting?

-        What is the added value?    

A graphical summary will help to disseminate the message.

Author Response

Point 1:

-        How can be translated to clinical setting?

-        What is the added value?    

We thank the reviewer for these remarks. We reformulated the last part of the discussion to answer these important point (line 310 to 325).

Point2 : A graphical summary will help to disseminate the message.

We agree with the reviewer’s comment. We created a graphical abstract that we submitted as a supplementary file.